# Platelet-Rich Plasma as Adjuvant Therapy for Recurrent Vesicovaginal Fistula: A Prospective Case Series

**DOI:** 10.3390/jcm8122122

**Published:** 2019-12-02

**Authors:** Dominika Streit-Ciećkiewicz, Konrad Futyma, Paweł Miotła, Magdalena Emilia Grzybowska, Tomasz Rechberger

**Affiliations:** 12nd Department of Gynecology, Medical University in Lublin, Jaczewskiego 8, 20-954 Lublin, Poland; 2Department of Gynecology, Gynecologic Oncology and Gynecologic Endocrinology, Medical University of Gdańsk, Ul. Smoluchowskiego 17, 80-214 Gdańsk, Poland

**Keywords:** platelet-rich plasma, vesicovaginal fistula, urogenital fistula, urinary diversion, fibrin glue, Latzko procedure

## Abstract

Vesicovaginal fistula (VVF) is the nonphysiological communication between the bladder and vagina, and surgical closure is the gold treatment standard. Despite that successful closure occurs in around 85% of patients after the first repair, recurrence remains a highly distressing complication for patients and surgeons. The aim of our study was to evaluate the efficacy of a platelet-rich plasma (PRP) injection as a supportive treatment in the surgical repair of recurrent VVF. Between January 2018 and July 2019, 16 patients with recurrent VVF were injected with PRP in a tertiary gynecological department. Subsequently, a surgical Latzko procedure for VVF closure was scheduled 6–8 weeks after the PRP injection allowing proper neovascularization and remodeling of surrounding tissues. Patients were considered cured if no leakage was observed after surgery and negative dye test results were indicated at follow-up. All patients who were examined therein remained dry. To the best of our knowledge, this is the first study aiming to assess PRP injections as a supporting treatment prior to surgical procedure for recurrent VVF. Preliminary results are encouraging, and we incorporated this method in our clinical practice. Further reports on a larger group will follow.

## 1. Introduction

Platelet-rich plasma (PRP) is an example of a new therapeutic approach for an expanding list of medical conditions. Beyond its established role in esthetic and sports medicine, PRP is being applied in other disciplines, as it has been found to enhance the tissue regenerative and repairing capacity. In PRP therapy, growth factors (GFs), released from platelets α-granules, accelerate stages of healing processes and bring about tissue necrosis resolution, chemotaxis, cell regeneration, cell proliferation and migration, extracellular matrix synthesis, remodeling, angiogenesis, and epithelialization [1]. In addition, PRP contains a high level of adhesion proteins, like fibrin, fibronectin, and vitronectin, that are the components of the extracellular matrix and play an important role in wound healing. Moreover, GFs stimulate neovascularization in healing sites by increasing the amount of nutrients needed for the attracted stem cells regenerating the damaged tissues. PRP by definition is a platelet plasma suspension of at least 200,000 platelets/µL [2]. In order to obtain PRP, whole venous blood with an anticoagulant is centrifuged and separated into different fractions containing appropriate ingredients. Of note, various marketed systems differ in the platelet concentration efficiency, leukocyte count, and ease of use [3]. Thus, classification of PRP, based on cell-type content (i.e., platelets, leukocytes) and fibrin density, was proposed by Dohan Ehrenfest et al. and divides the platelet-based concentrates into four types, shown in Table 1 [4].

Interestingly, it is hypothesized that the GFs released from platelets accelerate the integration of biologic grafts and healing. This may play an important role in new prostheses developments in the field of urogynecology. After implantation of the synthetic material, a chronic inflammatory reaction can occur with further transition into a foreign body reaction. Herein, fibroblasts and macrophages are responsible for the tissue response of either bio-tolerance or rejection of the foreign material during the healing phase [5,6]. In the study published by Gorlero et al., a sprayed form of an autologous platelet-rich fibrin (PRF) matrix was used as a supporting treatment in 10 women with recurrent pelvic organ prolapse (POP). Results after 24 months of follow-up were very encouraging, as the anatomical success rate was 80%, and prolapse symptoms improved by 100% in all patients [7].

Vesicovaginal fistula (VVF) is the nonphysiological communication between the bladder and vagina, resulting in uncontrollable urine leakage into the vagina. The most common causes of VVF in developed countries are gynecological and obstetric procedures. It is estimated that 85% of all VVFs appear as a complication of transabdominal hysterectomy (1.4/1000 procedures) or transvaginal hysterectomy (0.2/1000 procedures), and 11% develop after caesarean section. VVF can also be associated with uterine cavity curettage, cone biopsy, stress urinary incontinence procedures, or laparoscopic hysterectomy. The other, less common, factors are pelvic tumors, pelvic injuries, and foreign intrauterine or intravaginal bodies. VVF may also be a late consequence of oncological radiotherapy [8].

Surgical closure is the gold standard of VVF treatment, and according to the WHO report, the successful closure rate for a first attempt is 85%. Still, recurrence remains a highly distressing complication for patients and surgeons [9,10]. Since the first use of fibrin glue to close a postradiation VVF in a 43-year-old woman was published by Pettersson at al. [11] in 1979, several methods and different plasma concentrates have been used as salvage treatment for urogenital fistulas [10,12,13,14,15]. Of relevance to our study, Shirvan et al. reported the outcomes of VVF treatment only with PRP injection and platelet-rich fibrin glue without any other additional surgical procedure. Accordingly, 11 out of 12 patients with primary fistula became dry after 10 to 20 d of catheterization, depending on the diameter of the VVF [10].

The aim of our study was to evaluate the efficacy of a PRP injection as a supportive agent in the treatment of recurrent VVF after at least one previous unsuccessful attempt to close the fistula with conservative surgical treatment.

## 2. Materials and Methods

This prospective case series was conducted between January 2018 and July 2019. Sixteen patients with recurrent VVF were injected with PRP in a tertiary gynecological clinic and then underwent surgery. The inclusion criterion was recurrent VVF after at least one previous failed surgical attempt. Patients after supplementary radio-chemotherapy were not included in the study group. The diameter of the fistula was not part of the inclusion/exclusion criteria. The type of fistula was categorized according to the Goh classification [16]. All patients signed informed consent for supportive use of PRP and agreed to use their data for scientific purposes. The Local Ethics Committee approved the study concept (KE-0254/363/2018). The demographics of the patients are given in Table 2. 

In undertaking the procedure, whole blood (150–180 mL) was collected from the patients into sodium citrate tubes (ratio 9:1). The tubes were centrifuged with an Arthrex Angel System® kit (Arthrex Inc., Naples, USA), resulting in PRP volumes of 4–6 mL (Table 2).

### 2.1. PRP Injection

With the patient in the lithotomy position, the exact location of the fistula was determined. The PRP was injected transvaginally in 15 patients (Figure 1A,B) with a Secalon-T™ 130 mm needle (Merit Medical, South Jordan, UT, USA). 

In one patient, the injection was made via cystoscopy (Figure 2A–C) with an injeTAK^®^ adjustable tip cystoscopy needle (Laborie Medical Technologies, Canada) because of a narrow and long vagina with the fistula located on the anterior vaginal wall close to the vaginal apex. In all cases, PRP was injected at 4 to 5 points around the edges of the fistula.

After the injection, patients were discharged home with ciprofloxacin 500 mg, bid, for five days. The following surgical procedure for VVF closure was scheduled 6–8 weeks after the PRP injection to allow for proper neovascularization and remodeling of surrounding tissues.

### 2.2. VVF Repair Procedure

The Latzko procedure was performed with the patient in the lithotomy position while under general anesthesia. Surgery started with cystoscopy, ureters were localized, and single J catheters were inserted in order to decrease the amount of urine flowing into the bladder. The fistula was then visualized from the vagina, and a 6 or 10 Fr catheter (Balton Sp. Z o. o., Warszawa, Poland), depending on fistula size, was placed into the bladder via the VVF tract, and the balloon was inflated with 0.9% saline solution. The vaginal wall was then dissected and separated from the VVF tract approximately 1 cm around the fistula. The scar tissue of the fistulous tract was excised in order to refresh the edges for better healing. Subsequently, an imbricating three-layer closure of the bladder, vesicovaginal fascia, and vagina was done with the bladder wall being closed with 3–0 fast absorbable sutures (Monosyn Quick, Braun Surgical, Rubi, Spain). After the first layer, the suture tightness of the closure was checked by instilling 150 mL of methylene blue dye solution into the bladder. If watertight closure was achieved, the next two layers of 2–0 sutures (Novosyn, Braun Surgical, Rubi, Spain) were applied to completely secure the closure. A two-way DuFour catheter 20 Fr (Coloplast, Humlebæk, Denmark) and single J catheters were left for 7 d after surgery in order to decrease urine inflow into the bladder and to minimize the risk of bladder distension. Antibiotic prophylaxis (cefuroxime 1.5 g, tid) was administered during the catheterization period, while solifenacin 10 mg qd was given in order to decrease detrusor contractions. The single J catheters were then removed, along with the bladder catheter, while a Foley catheter of 18 Fr (Covidien, Malaysia) was kept for additional 7 d. After a total of 14 d, patient status was ascertained by releasing 150 mL methylene blue dye solution into the bladder before discharge. If no leakage was seen, the catheter was removed. The first follow-up visit was scheduled 4 to 6 weeks after discharge. Patients were recognized as cured if no leakage was observed during the follow-up period and after a negative dye test result at the follow-up visit. All patients are still being followed, and we have not observed recurrence of the VVF.

## 3. Results

Injection of PRP is a safe procedure, and we did not observe any complications or adverse reactions at the injection site. The Latzko procedure was performed in only 15 patients, as in one woman, the fistula self-healed 2 weeks after the PRP injection. The dates of the procedures are given in Table 3. All other patients were examined at the follow-up visit 4 to 6 weeks after the Latzko procedure and remained dry during that period. In all cases, the vaginal wall at the site of the procedure healed without any signs of scaring, redness, or granulosa tissue. Moreover, patients did not complain about any urination difficulties or urinary tract disorders. In addition, postvoid residuals were lower than 50 mL in all patients.

## 4. Discussion

Several studies have shown that the management of urogenital tract fistulas is a demanding procedure. Herein, the first attempt, either vaginal or abdominal, has the highest chance of success to restore proper bladder storage function and reverse urinary diversion through the vagina. This is because bladder wall healing proceeds in straitened circumstances, where urine is present on one side and vaginal flora on the other [17,18,19]. As we already mentioned, in Western countries, the majority of VVFs are commonly associated with iatrogenic injury that comes about during gynecological surgery, mainly hysterectomy [20,21]. Among many risk factors predisposing patients to fistulae in industrialized countries, pelvic surgery, radiation, prolonged presence of a foreign body, infection, and pelvic malignancy are most commonly encountered [22,23]. A recently published review concerning 5698 hysterectomies revealed that VVF formation after iatrogenic bladder injury during hysterectomy was associated with larger uteri, longer surgery time, and more severe bladder injuries [24]. Nevertheless, there is no doubt that vesicovaginal fistula formation, especially several days after index surgery, is caused from tissue ischemia due to traumatization during operation.

This is the main reason why additional healing-enhancing techniques should be involved in the treatment of recurrent VVF so as to improve the anatomical success rate and quality of life for the patient. PRP contains a concentration of platelets several times higher than that of the physiological norm and, thus, a higher concentration of growth factors localized in thrombocytes. Among these are platelet-derived growth factor (PDGF), transforming growth factor ß (TGF ß), vascular endothelial growth factor (VEGF), and endothelial growth factor (EGF) as well as adhesion proteins involved in tissue healing [1,5].

Our results are similar to previous studies concerning the treatment of VVF. In our study, all patients injected with PRP were healed and remained dry. Shirvan et al. [10] developed a primary procedure consisting of PRP injection around the fistula into the tissue supplemented with platelet-rich fibrin (PRF) glue interpositioned inside the fistulous canal. After 6 months of follow up, 11 out of 12 patients (92%) were clinically cured and had normal findings on transvaginal physical examinations and cystography. They concluded that autologous PRP with PRF glue interposition is a safe, effective, and minimally invasive approach for the treatment of VVF that obviated the need for surgery.

This study describes the effect of platelet-based concentrates on the treatment of VVF in the largest group of patients to date. In our study, we were unable to use the Hamidi–Shirvan method because our patients presented with recurrent fistula >5 mm in diameter, where scar tissue after previous attempts was overgrown and had a negative influence on wound healing. The key point before the surgical procedure in our patients was to restore the tissue’s ability to heal. Utilizing the PRP as a donor of platelets containing bioactive factors and cytokines is commonly accepted in regenerative medicine. We assumed that regeneration of tissues around the fistula will energize the healing processes after its surgical closure. To achieve proper neovascularization and prepare the surrounding tissues, we injected PRP six to eight weeks prior to surgery. In contrast, Morita and Tokue reported on the successful closure of a radiation-induced vesicovaginal fistula, 5 mm in diameter, wherein bovine collagen was injected during cystoscopy into the submucosal layer around the fistula after fulguration of the fistula edges. Fibrin glue was then interpositioned transvaginally into the fistulous tract and was retained in place after blocking the vaginal orifice of the fistula with a collagen injection [25]. This might be an option for small fistulas with a relatively long tract, but in case of posthysterectomy fistulas, the fistulous tract is usually very short, and the diameter is usually greater than 5 mm, as the scaring tissue shrinks after each previous surgery and pulls the edges of the fistula away from center. Those specific conditions hinder fibrin glue deposits from adhering to the surrounding tissue and enforcing the watertight seal.

There are several case reports or case series that describe the resolution of nonmalignant VVF after administration of fibrin sealant, but no further studies on larger groups are available [10,12,26,27,28].

To the best of our knowledge, this is the first study aimed at assessing injection of PRP as a supporting treatment prior to surgical procedure, which is still the leading method applied to close large, recurrent vesicovaginal fistulas. Preliminary results are encouraging, and we now have incorporated this method into our clinical practice. The limitation of this study is the relatively short observation time, but all patients are still being followed, and further reports on a larger group will follow.

## Figures and Tables

**Figure 1 jcm-08-02122-f001:**
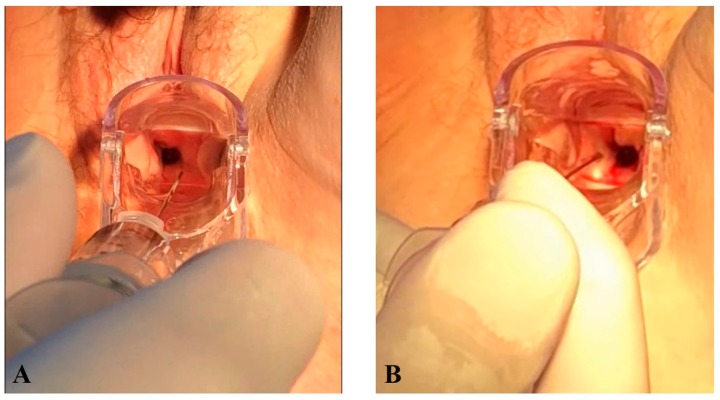
Transvaginal PRP injection around the edges of the fistula. (**A**,**B**) Transvaginal PRP injection around the edges of the fistula.

**Figure 2 jcm-08-02122-f002:**
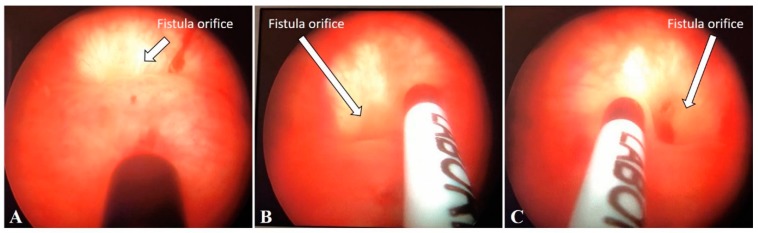
PRP injection via cystoscope. Arrows pointing the fistula bladder orifice. (**A**–**C**) PRP injection via cystoscope.

**Table 1 jcm-08-02122-t001:** Classification of platelet-based concentrates proposed by Dohan Ehrenfest et al. [4].

Preparation	Acronym	Leucocyte count	Fibrin density
Pure platelet-rich plasma	P-PRP	Poor	Low
Leukocyte- and platelet-rich plasma	L-PRP	Rich	Low
Pure platelet-rich fibrin	P-PRF	Poor	High
Leukocyte- and platelet-rich fibrin	L-PRF	Rich	High

**Table 2 jcm-08-02122-t002:** Demographic, clinical, and procedural data of the patients.

*n*	Age (years)	BMI (kg/m^2^)	Parity (Vaginal/CS)	Preceding Procedure	Goh’s Classification of VVF	Number of Previous Attempts of Surgical Closure Before PRP Injection	Injected PRP Volume (mL)
1	36	23.44	1/0	MEIGS	1aiii	1	5.0
2	47	25.51	1/0	TLH/BSO	1ai	1	5.0
3	74	29.74	4/0	TVH	1bii	1	6.0
4	41	42.81	4/1	TAH/BSO	3ai	1	5.0
5	38	31.25	1/0	TOT	4bii	2	6.0
6	55	29.37	0/1	TAH/BSO	2bi	2	5.0
7	47	28.8	0/2	TAH/BSO	2bii	1	3.0
8	41	24.22	3/1	TAH/BSO	1aiii	1	6.0
9	37	22.68	0/2	TAH/BSO	1ai	2	5.0
10	55	27.39	1/0	LSH/BSO	3ai	2	5.0
11	75	23.58	1/0	TAH/BSO	2bii	2	6.0
12	74	20.98	3/0	MEIGS	3aii	2	3.5
13	59	26.88	2/0	TAH/BSO	2ai	2	3.0
14	72	24.91	4/0	TAH/BSO	1aii	2	4.0
15	58	20.43	1/0	TAH	1aii	2	3.0
16	42	19.53	0/0	TAH	1aii	2	3.0

Legend: TLH/BSO—total laparoscopic hysterectomy and bilateral salpingoophorectomy; TAH/BSO—total abdominal hysterectomy and bilateral salpingoophorectomy; TVH—total vaginal hysterectomy; MEIGS—Wertheim–Meigs radical hysterectomy; TOT—transobturator tape; CS—caesarean section; VVF—vesicovaginal fistula; PRP—platelet-rich plasma; and BMI—body mass index.

**Table 3 jcm-08-02122-t003:** Dates of the procedures.

*N*	Previous surgery	PRP injection	Following surgery
1	06.10.2016	05.01.2018	
2	16.11.2017	08.01.2018	20.03.2018
3	18.12.2017	25.04.2018	22.06.2018
4	22.02.2018	26.04.2018	08.06.2018
5	29.03.2018	27.06.2018	10.08.2018
6	27.02.2018	03.09.2018	05.11.2018
7	30.07.2018	24.10.2018	03.12.2018
8	13.04.2018	16.11.2018	03.01.2019
9	23.04.2018	03.12.2018	18.01.2019
10	06.02.2017	19.12.2018	30.01.2019
11	09.05.2019	25.06.2019	01.08.2019
12	19.04.2019	26.06.2019	29.07.2019
13	04.04.2019	01.08.2019	16.09.2019
14	28.06.2019	13.08.2019	26.09.2019
15	17.07.2019	20.08.2019	04.10.2019
16	12.07.2019	28.08.2019	11.10.2019

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
