# Peer review of "Platelet-Rich Plasma as Adjuvant Therapy for Recurrent Vesicovaginal Fistula: A Prospective Case Series"

_jcm, 2019, doi:10.3390/jcm8122122_

Round 1

Reviewer 1 Report

The authors are to be congratulated for investigating a new technique designed to enhance the treatment of a debilitating condition in women.  This is a novel idea that deserves further research based on the results of this study.

I have the following comments:

1)  The prose within the manuscript could benefit from editing regarding proper English; it is apparent that English is not the first language of the authors.

2)  In the Methods section the authors should comment on the study design; it appears to be a prospective case series, and should be identified as such.

3)  The Methods would also benefit from a list on Inclusion and Exclusion criteria of the subjects.

4) in either the Methods, or preferably in the Results section, the authors should list the timing of the PRP injections and secondary repair in relation to the time of the primary repair.

5)  The follow-up time of 4-6 weeks is very limited and should be noted as a limitation of the study.  If possible, the study would benefit greatly form an update on success rates at longer follow-up.

6)  Was having a size of >5mm of the recurrent fistula (as noted in line 162 of the Discussion section an inclusion criteria, or wa this just a coincidence that all the cases had a fistula of this size?

Author Response

The authors are to be congratulated for investigating a new technique designed to enhance the treatment of a debilitating condition in women.  This is a novel idea that deserves further research based on the results of this study.

I have the following comments:

1)  The prose within the manuscript could benefit from editing regarding proper English; it is apparent that English is not the first language of the authors.

The manuscript was checked before initial submission by native speaker. We have send it again for English editing and all changes are highlighted in the manuscript.

2)  In the Methods section the authors should comment on the study design; it appears to be a prospective case series, and should be identified as such.

We have added the information about the type of the study in the title and in the Methods section.

3)  The Methods would also benefit from a list on Inclusion and Exclusion criteria of the subjects.

We have added this information in the body text.

4) in either the Methods, or preferably in the Results section, the authors should list the timing of the PRP injections and secondary repair in relation to the time of the primary repair.

Those information were added in the Table 3 in the results section.

5)  The follow-up time of 4-6 weeks is very limited and should be noted as a limitation of the study.  If possible, the study would benefit greatly form an update on success rates at longer follow-up.

The limitation of the study was added in the Discussion section. But we have those patients under the supervision and all of them are still dry, even after almost 19 months. For the purposes of this article we chose to present the results after 4-6 weeks because it allowed us to include more patients. We are planning to publish our results based on the group of at least 40 cases with much longer follow-up period but assume that we will be able to do this in two years.

6)  Was having a size of >5mm of the recurrent fistula (as noted in line 162 of the Discussion section an inclusion criteria, or wa this just a coincidence that all the cases had a fistula of this size?

It was a coincidence and we added the information that diameter of the fistula was not  limiting inclusion or exclusion criteria in Materials and methods.

Reviewer 2 Report

This is a very interesting study done on an under-studied population. I commend the authors for trying a new technology for this problem. I would like to see this manuscript published. 

The only weakness I detect is the lack of a control group. If the authors have any data on previous cases or other cases done in their institution of recurrent fistulas that were repaired without PRP, it would be helpful to cite the success/failure rate of those cases. 

Author Response

This is a very interesting study done on an under-studied population. I commend the authors for trying a new technology for this problem. I would like to see this manuscript published. 

The only weakness I detect is the lack of a control group. If the authors have any data on previous cases or other cases done in their institution of recurrent fistulas that were repaired without PRP, it would be helpful to cite the success/failure rate of those cases. 

We are aware that we do not have the control group but in our opinion every patient is a control for himself, because previous surgery failed to close the fistula. Closure rate of primary procedure, without PRP in our department is similar to that published in the literature. Moreover, only 4 patients from this group had primary procedure done in our department, the remaining patients were referred from different centers. That is why the success rate of primary closure rate in our department would not be a referential for the study group.